# Cal-DETR: Calibrated Detection Transformer

**Muhammad Akhtar Munir**[1,2] , **Salman Khan**[1,3]**, Muhammad Haris Khan**[1],
**Mohsen Ali**[2]**, Fahad Shahbaz Khan**[1,4]
[1]Mohamed bin Zayed University of AI, [2]Information Technology University,
[3]Australian National University, [4]Linköping University
`akhtar.munir@mbzuai.ac.ae`

## Abstract

Albeit revealing impressive predictive performance for several computer vision tasks, deep neural networks (DNNs) are prone to making overconfident predictions. This limits the adoption and wider utilization of DNNs in many safety-critical applications. There have been recent efforts toward calibrating DNNs, however, almost all of them focus on the classification task. Surprisingly, very little attention has been devoted to calibrating modern DNN-based object detectors, especially detection transformers, which have recently demonstrated promising detection performance and are influential in many decision-making systems. In this work, we address the problem by proposing a mechanism for calibrated detection transformers (Cal-DETR), particularly for Deformable-DETR, UP-DETR and DINO. We pursue the train-time calibration route and make the following contributions. *First*, we propose a simple yet effective approach for quantifying uncertainty in transformer-based object detectors. *Second*, we develop an uncertainty-guided logit modulation mechanism that leverages the uncertainty to modulate the class logits. *Third*, we develop a logit mixing approach that acts as a regularizer with detection-specific losses and is also complementary to the uncertainty-guided logit modulation technique to further improve the calibration performance. Lastly, we conduct extensive experiments across three in-domain and four out-domain scenarios. Results corroborate the effectiveness of Cal-DETR against the competing train-time methods in calibrating both in-domain and out-domain detections while maintaining or even improving the detection performance. Our codebase and pre-trained models can be accessed at `https://github.com/akhtarvision/cal-detr`.

## 1 Introduction

Deep neural networks (DNNs) have enabled notable advancements in computer vision, encompassing tasks such as classification [29, 6, 44, 11], object detection [2, 53, 5, 38, 43, 51], and semantic segmentation [47, 42, 3]. Due to their high predictive performance, DNNs are increasingly adopted in many real-world applications. However, it has been shown that DNNs often make highly confident predictions even when they are incorrect [10]. This misalignment between the predictive confidence and the actual likelihood of correctness undermines trust in DNNs and can lead to irreversible harm in safety-critical applications, such as medical diagnosis [7, 41], autonomous driving [9], and legal decision-making [49].

Among others, an important factor behind the overconfident predictions of DNNs is the training with zero-entropy target vectors, which do not encompass uncertainty information [12]. An early class of methods aimed at reducing the model's miscalibration proposes various post-hoc techniques [10, 15, 16, 37, 21]. Typically, a single temperature scaling parameter is used to re-scale the logits of a trained model which is learned using a held-out validation set. Despite being effective and simple, they are architecture-dependent and require a held-out set which is unavailable in many real-world scenarios [28]. Recently, we have seen an emergence of train-time calibration methods. They are

typically based on an auxiliary loss that acts as a regularizer for predictions during the training process [12, 28, 25, 19, 33]. For instance, [28] imposes a margin constraint on the logit distances in the label smoothing technique, and [12] develops an auxiliary loss that calibrates the confidence of both the maximum class and the non-maximum classes. These train-time approaches reveal improved calibration performance than various existing post-hoc methods.

We note that majority of the DNN calibration methods target the classification task [10, 15, 16, 37, 21, 12, 28, 25, 19]. Strikingly, much little attention has been directed towards studying the calibration of object detectors [33]. Object detection is a fundamental task in computer vision whose application to safety-critical tasks is growing rapidly. Furthermore, most of the existing methods focus on calibrating in-domain predictions. However, a deployed model could be exposed to constantly changing distributions which could be radically different from its training distribution. Therefore, in this paper, we aim to investigate the calibration of object detection methods both for *in-domain* and *out-domain* predictions. Particularly, we consider the recent transformer-based object detection methods [5, 53, 51] as they are the current state-of-the-art with a simpler detection pipeline. Moreover, they are gaining increasing attention from both the scientific community and industry practitioners.

In this paper, inspired by the train-time calibration paradigm, we propose an uncertainty-guided logit modulation and a logit mixing approach to improve the calibration of detection transformers (dubbed as Cal-DETR). In order to modulate the logits, we propose a new and simple approach to quantify uncertainty in transformer-based object detectors. Unlike existing approaches [8, 23], it does not require any architectural modifications and imposes minimal computational overhead. The estimated uncertainty is then leveraged to modulate the class logits. Next, we develop a mixing approach in the logit space which acts as a regularizer with detection-specific losses to further improve the calibration performance. Both uncertainty-guided logit modulation and logit mixing as a regularizer are complementary to each other. We perform extensive experiments with three recent transformer-based detectors (Deformable-DETR (D-DETR) [53], UP-DETR [5] and DINO [51]) on three in-domains and four out-domain scenarios, including the large-scale object detection MS-COCO dataset [27]. Our results show the efficacy of Cal-DETR against the established train-time calibration methods in improving both in-domain and out-domain calibration while maintaining or even improving detection performance.

## 2   Related Work

**Vision transformer-based object detection:** Vision transformers are enabling new state-of-the-art performances for image classification [6, 46], semantic segmentation [47, 42], and object detection [2, 53, 5, 51]. The work of DETR [2] introduced the first-ever transformer-based object detection pipeline to the vision community by eliminating post-processing steps like non-maximum suppression with object matching to particular ground truth using bipartite matching. However, DETR is very slow in convergence and performs poorly on small objects, and to address these issues, Deformable-DETR [53] was proposed that is based on a deformable attention module as opposed to vanilla self-attention in DETR [2]. The work of [5] further enhanced the performance of this object detection pipeline by pretraining the encoder-decoder parameters. Recently, DINO [51] improved detection performance with contrastive denoising training, look forward twice approach and integrated effective initialization of anchors.

**Model calibration:** Model calibration requires the perfect alignment of model's predictive confidence with the actual likelihood of its correctness. One of the earliest techniques for model calibration is known as post-hoc temperature scaling (TS) [10], which is a variant of Platt scaling [37]. The idea is to re-scale the logits with a single temperature parameter (T) which is learned using a held-out validation set. However, a limitation of TS is that the selection of temperature parameter T value is reliant on the network architecture and data [28]. And, TS has been shown to be effective for calibrating in-domain predictions. To extend TS to domain-drift scenarios, [45] proposed to perturb the held-out validation set before the post-hoc processing step. Dirichlet calibration (DC) [20] was extended from beta calibration [21] to a multi-class setting. It uses an extra neural network layer that log-transforms the miscalibrated probabilities, and the parameters for this layer are learned using the validation set. Despite being effective and computationally inexpensive, post-hoc calibration methods demand the availability of a held-out validation set, which is a difficult requirement to be met for many real-world use cases.

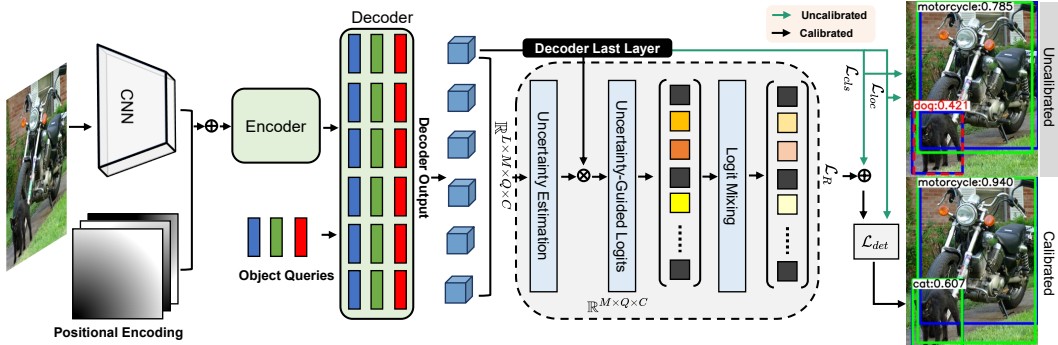

Figure 1: *Main architecture:* An image is passed through a feature extractor CNN and unmodified transformer's encoder structure, whereas the decoder is modified to get the model well calibrated. Logits are modulated based on uncertainty quantification that exploits the inbuilt design of the decoder without extra computational cost (Sec. 3.3.1). This module is followed by a logit mixing mechanism which is comprised of a regularizer ($\mathcal{L}_R$) to obtain calibrated predictions (Sec. 3.3.2). *<category:confidence>* are depicted on the right side of figure. The baseline detector (D-DETR [53]) outputs wrong *dog* prediction with a relatively high confidence, however, our Cal-DETR provides accurate *cat* prediction with a high confidence. Green boxes are accurate predictions whereas Red (dashed) box is inaccurate prediction. Blue bounding boxes represent ground truths for corresponding detections. $\mathcal{L}_{cls}$ and $\mathcal{L}_{loc}$ are classification and localization losses respectively [5, 53, 51].

Train-time calibration methods tend to engage model parameters during training by proposing an auxiliary loss function that is optimized along with the task-specific loss function [12, 28]. Some methods aim at maximizing the entropy of the predictive distribution either explicitly or implicitly. For instance, [35] formulated an auxiliary term, that is based on negative of entropy, which is minimized during training to alleviate the impact of overconfident predictions. Along similar lines, [32] investigated that the label smoothing (LS) technique is also capable of reducing miscalibration, and [31] demonstrated that the focal loss (FL) can improve the model calibration. These methods are effective, however, their logit-distance constraints can often lead to non-informative solutions. To circumvent this, [28] proposed a margin-based generalization of logit-distance constraints to achieve a better balance between discriminative and calibration performance. [12] proposed to calibrate the confidence of both predicted and non-predicted classes by formulating an auxiliary loss term known as the multi-class difference of confidence and accuracy. All aforementioned calibration techniques target classification task, recently [33] developed a train-time calibration loss for object detection which imposes constraints on multi-class predictions and the difference between the predicted confidence and the IoU score.

**Uncertainty estimation and model calibration:** Owing to the computational intractability of Bayesian inference, different approximate inference methods have been proposed, such as variational inference [1, 30], and stochastic expectation propagation [14]. The work of [34] evaluated predictive uncertainty of models under distributional shift and investigates its impact on the accuracy-uncertainty trade-off. [19] leveraged the predictive uncertainty of model and proposed to align this with accuracy, and [18] induced soft bins and makes calibration error estimates differentiable. It is also possible to quantify uncertainty by ensemble learning that takes into account empirical variance of model predictions. It allows better discriminative performance and improved predictive uncertainty estimates leading to well-calibrated models. Ensembles can emerge with difference in weights initializations and random shuffling of training examples [23] and Monte Carlo (MC) dropout [8]. However, ensemble-based uncertainty estimation techniques could be computationally expensive, especially with complex models and large-scale datasets. In this work, we propose a simple and computationally efficient technique to quantify uncertainty in a transformer-based object detection pipeline.

# 3 Method

This section begins by discussing the preliminaries, including the problem settings and the background knowledge relevant to calibrating object detectors. We then detail our proposed approach, Cal-DETR, encompassing uncertainty-guided logit modulation and logit mixing for calibrating the recent vision transformer-based (ViT) object detection methods (Fig. 1).

### 3.1 Preliminaries

Assume a joint distribution $\mathcal{S}(\mathcal{X}, \mathcal{Y})$, where $\mathcal{X}$ is an input space and $\mathcal{Y}$ is the corresponding label space. Let $\mathcal{D} = \{(\mathbf{x}_i, \hat{\mathbf{y}}_i)_{i=1}^N\}$ be the dataset sampled from this joint distribution consisting of pairs of input image $\mathbf{x}_i \in \mathcal{X}$ and corresponding label $\hat{\mathbf{y}}_i \in \mathcal{Y}$. The input image belongs to $\mathbf{x}_i \in \mathbb{R}^{H \times W \times Z}$, where $H$ is the height, $W$ is the width and $Z$ is the number of channels. For each input image $\mathbf{x}_i$, we have the corresponding ground truth label, $\hat{\mathbf{y}}_i \in \mathbb{R}^5$, and it consists of class label and the bounding box coordinates $(\hat{c}, \hat{x}, \hat{y}, \hat{w}, \hat{h})$. Here $\hat{c}$ is the class label $\in \{1, 2, ..., C\}$ and $\hat{x}, \hat{y}, \hat{w}, \hat{h}$ are the bounding box $\hat{\mathbf{b}} \in \mathbb{R}^4$ coordinates. Next, we define the calibration of a model in classification and object detection tasks.

A model is said to be calibrated when its accuracy perfectly aligns with the predictive confidence. An uncalibrated model is prone to making over-confident and under-confident predictions. To mitigate these problems, for an image, a model should be able to predict the class confidence that matches with the actual likelihood of its correctness. Let $\mathcal{H}_{cls}$ be a classification model which predicts a class label $\bar{c}$ with the confidence score $\bar{z}$. Now, the calibration for the classification task can be defined as: $\mathbb{P}(\bar{c} = \hat{c} | \bar{z} = z) = z$, s.t., $z \in [0, 1]$ [22]. This expression shows that, to achieve the perfect calibration in classification, the predicted confidence and accuracy must match.

Let $\mathcal{H}_{det}$ be the detection model, which predicts a bounding box $\bar{\mathbf{b}}$ and the corresponding class label $\bar{c}$ with the confidence score $\bar{z}$. The calibration for a detection task is defined as $\mathbb{P}(F = 1 | \bar{z} = z) = z$, $\forall z \in [0, 1]$ [22][1]. Where $F = 1$ denotes the true prediction of the object detection model which occurs when the predicted and the ground truth classes are the same, and the Intersection over Union (IoU) of both the predicted and ground truth boxes are above a pre-defined threshold $k$ i.e. $\mathbb{1}[IoU(\bar{\mathbf{b}}, \hat{\mathbf{b}}) \geq k] \mathbb{1}[\bar{c} = \hat{c}]$. $F = 0$ denotes the false prediction and together with $F = 1$, the precision is determined. In object detection, to achieve perfect calibration, the precision of a model must match the predicted confidence. In the next subsection, we revisit the formulae for measuring calibration.

### 3.2 Measuring Calibration

**Classification:** For classification problems, miscalibration is measured by computing Expected Calibration Error (ECE) [10]. It computes the expected divergence of accuracy from all possible confidence levels as: $\mathbb{E}_{\bar{z}}[|\mathbb{P}(\bar{c} = \hat{c} | \bar{z} = z) - z|]$. In practice, the ECE metric is approximated by (1) first dividing the space of confidence scores into B equal-sized bins, (2) then computing the (absolute) difference between the average accuracy and the average confidence of samples falling into a bin, and finally (3) summing all the differences after weighting each difference by the relative frequency of samples in a bin.

$$\text{ECE} = \sum_{b=1}^B \frac{|S(b)|}{|D|} \left| \text{accuracy}(b) - \text{confidence}(b) \right|, \tag{1}$$

where, $\text{accuracy}(b)$ and $\text{confidence}(b)$ represent the average accuracy and the average confidence of samples in the $b^{th}$ bin, respectively. $S(b)$ is a set of samples in the $b^{th}$ bin and $|D|$ is the total number of samples.

**Object Detection:** Similar to ECE, the Detection Expected Calibration Error (D-ECE) is measured as the expected divergence of precision from confidence, at all confidence levels [22]. It can be expressed as $\mathbb{E}_{\bar{s}}[|\mathbb{P}(F = 1 | \bar{z} = z) - z|]$. As in Eq.(1), confidence is a continuous variable, therefore we can approximate D-ECE:

$$\text{D-ECE} = \sum_{b=1}^B \frac{|S(b)|}{|D|} \left| \text{precision}(b) - \text{confidence}(b) \right|, \tag{2}$$

where $S(b)$ contains a set of object instances in $b^{th}$ bin. Note that, the D-ECE considers only the precision vs. confidence, and not the mean Average Precision (mAP). The mAP requires the computation of recall for which detections are not present, and so there are no associated confidences.

---

[1]This calibration definition of object detectors can be extended after including box information.

### 3.3 Cal-DETR: Uncertainty-Guided Logit Modulation & Mixing

Modern object detection methods occupy a central place in many safety-critical applications, so it is of paramount importance to study their calibration. Recently, vision transformer-based (ViT) object detectors (e.g., Deformable-DETR (D-DETR) [53], UP-DETR [5] and DINO [51]) are gaining importance for both industry practitioners and researchers owing to their state-of-the-art performance and relatively simple detection pipeline. Therefore, we aim to study the calibration of ViT-based object detectors. Towards improving calibration in transformer-based object detectors, we first describe our uncertainty-guided logit modulation approach (Sec. 3.3.1) and then propose a new logit mixing regularization scheme (Sec.3.3.2).

#### 3.3.1 Uncertainty-Guided Logit Modulation

We first briefly review the DETR pipeline [2], which is a popular ViT-based object detection pipeline. Recent transformer-based detectors [5, 53, 51] largely borrow the architecture of DETR pipeline [2]. Then, we describe our new uncertainty quantification approach and how it is leveraged to modulate the logits from the decoder to eventually calibrate confidences.

**Revisiting DETR pipeline:** DETR [2] was proposed as the first ViT-based object detection pipeline with the motivation of eliminating the need for hand-designed components in existing object detection pipelines. It consists of a transformer encoder-decoder architecture with a set-based Hungarian loss which ensures unique predictions against each ground truth bounding box. Assuming the existence of feature maps $f \in \mathbb{R}^{H \times W \times C}$ which are generated from ResNet [11] backbone, DETR relies on the standard transformer encoder-decoder architecture to project these feature maps as the features of a set of object queries. A feed-forward neural network (FFN) in a detection head regresses the bounding box coordinates and the classification confidence scores.

**Quantifying uncertainty in DETR pipeline:** Some recent studies have shown that the overall (output) predictive uncertainty of the model is related to its calibration [23, 19]. So estimating a model's uncertainty could be helpful in aiding the confidence predictions to align with the precision of a detection model. To this end, we propose a new uncertainty quantification approach for DETR pipeline after leveraging the innate core component of transformer architecture i.e. the decoder. It is a simple technique that requires no changes to the existing architecture and yields minimal computational cost, unlike Monte Carlo dropout [8] and Deep Ensembles [23]. Specifically, let $O_D \in \mathbb{R}^{L \times M \times Q \times C}$ and $O_D^L \in \mathbb{R}^{M \times Q \times C}$ be the outputs of all decoder layers and final decoder layer of the detection transformer, respectively. Where $L$ is the number of decoder layers, $M$ is the mini-batch, $Q$ is the number of queries, and $C$ represents the number of classes. To quantify the uncertainty for every class logit $u_c$, we compute the variance along the $L$ dimension which consists of multiple outputs of decoder layers. In other words, the variation in the output of decoder layers reflects the uncertainty for a class logit $u_c$. Overall, after computing variance along the $L^{th}$ dimension in $O_D$, we obtain a tensor $\mathbf{u} \in \mathbb{R}^{M \times Q \times C}$, which reflects the uncertainty for every class logit across query and minibatch dimensions.

**Logit modulation:** We convert the uncertainty $\mathbf{u}$ into certainty using: $1 - \tanh(\mathbf{u})$, where $\tanh$ is used to scale the uncertainty values to $[0, 1]$ from $[0, \inf]$, and use this certainty to modulate the logits output from the final layer as:

$$\tilde{O}_D^L = O_D^L \otimes (1 - \tanh(\mathbf{u})), \tag{3}$$

where $\tilde{O}_D^L$ is the uncertainty-modulated decoder output and then this becomes the input to the loss function. $1 - \tanh(\mathbf{u})$ indicates that if an output in logit space shows higher uncertainty, it will be down-scaled accordingly. In other words, a higher uncertainty implicitly implies a likely poor calibration of the output logit value (i.e. the pre-confidence score).

#### 3.3.2 Logit Mixing for Confidence Calibration

In addition to uncertainty-guided logit modulation, we develop a new logit mixing approach which acts as a regularizer with the detection-specific losses for further improving the confidence calibration. We propose logit mixing as the mixing in the logit space of an object detector which is inspired by the input space mixup for classification [50]. In a DETR pipeline, this logit space is of dimensions $Q \times C$, where $Q$ is the number of object queries and $C$ is the number of classes. We only consider positive queries for performing logit mixing as they are responsible for the detection of objects. In

object detection, the instance-level object information is important as we consider the confidence scores associated with the positive class assigned to the bounding boxes.

In our logit mixing scheme, we first build a prototypical representation by computing the mean across all positive queries, which is then used to achieve the mixing for any given positive query. For example, given three positive queries, let $Q_p = \{Q_1, Q_2, Q_3\} \subset Q$ (total number of queries) belonging to the object classes and let $\tilde{Q}$ be a prototypical representation of all the positive queries, we formulate the expression as:

$$\acute{Q}_i = \alpha Q_i + (1 - \alpha)\tilde{Q}. \qquad \forall\, Q_i \in Q_p \tag{4}$$

Now, $\{\acute{Q}_1, \acute{Q}_2, \acute{Q}_3\}$ are the mixed versions of $\{Q_1, Q_2, Q_3\}$ in the logit space. $\alpha$ is the mixing weight for logits. For the selection of $\alpha$, more details are in Sec. 4.2.

We employ a regularization loss by leveraging the proposed logit mixing output, denoted as $\acute{\psi}(\mathbf{x}_i)$, which is expressed as: $\mathcal{L}_R(p(\bar{c}|\acute{\psi}(x_i), \acute{c}_i)$, where $\acute{c}_i$ contains the labels after smoothing with mixing values. Instead of one-hot labels, $\alpha$ determines the weightage of classes for object instances. Specifically $\alpha$ value for a given positive query contributes to smoothing the label while remaining positive queries that contribute to forming a prototypical representation share $1$-$\alpha$ value to smooth corresponding labels uniformly. The loss $\mathcal{L}_R$ is used together with the task-specific classification loss $\mathcal{L}_{cls}(p(\bar{c}|\psi(x_i)), \hat{c}_i)$ to obtain a joint loss formulation as: $\mathcal{L}_{cls} + \lambda\mathcal{L}_R$, where $\psi(\mathbf{x}_i)$ denotes the non-mixed logits and $\lambda$ controls the contribution of regularization loss. The proposed regularizer loss tends to capture the information from the vicinity of object instances in the logit space by mixing logits and constructing a non-zero entropy target distribution. This penalizes the model for producing overconfident predictions by possibly reducing overfitting [52, 36]. Our empirical results (Sec. 4.1) show that this not only improves model calibration but also improves the overall detection performance in most scenarios and it is also complementary to our uncertainty-guided logit modulation technique.

## 4    Experiments & Results

**Datasets:** To perform experiments, we utilize various in-domain and out-domain benchmark datasets. Details are as follows: **MS-COCO** [27] is a large-scale object detection dataset for real images containing 80 classes. It splits into 118K train images, 41K test images, and 5K validation images. The train set (train2017) is used for training while the validation set (val2017) is utilized for evaluation. **CorCOCO** contains a similar set of images as present in val2017 of MS-COCO but with the corrupted version [13]. Random corruptions with random severity levels are introduced for evaluation in an out-domain scenario. **Cityscapes** [4] is an urban driving scene dataset contains 8 classes *person, rider, car, truck, bus, train, motorbike, and bicycle*. The training set consists of 2975 images while the 500 validation images are used as the evaluation set. **Foggy Cityscapes** [39] is a foggy version of Cityscapes, 500 validation images simulated with severe fog are used for evaluation in this out-domain scenario. **BDD100k** [48] contains 100k images from which 70K are training images, 20K are for the test set and 10K for the validation set. Categories are similar to Cityscapes and for evaluation, we consider the validation set and make a subset of daylight images as an out-domain scenario, reducing the set to 5.2K images. **Sim10k** [17] contains 10k synthetic images with car category from which the training split is 8K images and for the evaluation set, the split is 1K images.

**Datasets (validation post-hoc):** For the given scenarios, we opt separate validation datasets for the post-hoc calibration method (temperature scaling). In the MS-COCO scenario, we utilize the Object365 [40] validation dataset reflecting similar categories. For the Cityscapes scenario, a subset of the BDD100k train set is used as a validation set. And for the Sim10k scenario, we use the remaining split of images.

**Detection Calibration Metric:** For the calibration of object detectors, we report our results with the evaluation metric, which is the detection expected calibration error (D-ECE) [22]. It is computed for both in-domain and out-domain scenarios.

**Implementation Details:** We use transformer-based state-of-art-detectors Deformable-DETR (D-DETR), UP-DETR, and DINO as baselines respectively, which are uncalibrated. With the respective in-domain training datasets, we train D-DETR with our proposed method and compare it with baseline, post-hoc approach [10], and train time calibration losses [28, 12, 33] integrated with D-DETR. We

| Datasets / Methods | In-Domain (COCO) | | | Out-Domain (CorCOCO) | | |
|---|---|---|---|---|---|---|
| | D-ECE $\downarrow$ | AP $\uparrow$ | $AP_{0.5}$ $\uparrow$ | D-ECE $\downarrow$ | AP $\uparrow$ | $AP_{0.5}$ $\uparrow$ |
| Baseline $_{D\text{-}DETR}$ [53] | 12.8 | 44.0 | 62.9 | 10.8 | 23.9 | 35.8 |
| Temp. Scaling [10] | 14.2 | 44.0 | 62.9 | 12.3 | 23.9 | 35.8 |
| MDCA [12] | 12.2 | 44.0 | 62.9 | 11.1 | 23.5 | 35.3 |
| MbLS [28] | 15.7 | 44.4 | 63.4 | 12.4 | 23.5 | 35.3 |
| TCD [33] | 11.8 | 44.1 | 62.9 | 10.4 | 23.8 | 35.6 |
| Cal-DETR (Ours) | 8.4 | 44.4 | 64.2 | 8.9 | 24.0 | 36.3 |

Table 1: *Deformable-DETR (D-DETR) Calibration*: MS-COCO & Cor-COCO. Calibration results along with the detection performance. Cal-DETR improves calibration as compared to baseline, other train-time losses, and TS methods.

| Datasets / Methods | In-Domain (COCO) | | | Out-Domain (CorCOCO) | | |
|---|---|---|---|---|---|---|
| | D-ECE $\downarrow$ | AP $\uparrow$ | $AP_{0.5}$ $\uparrow$ | D-ECE $\downarrow$ | AP $\uparrow$ | $AP_{0.5}$ $\uparrow$ |
| Baseline $_{UP\text{-}DETR}$ [5] | 25.5 | 37.1 | 57.3 | 27.5 | 19.9 | 32.5 |
| Temp. Scaling [10] | 26.4 | 37.1 | 57.3 | 28.2 | 19.9 | 32.5 |
| MDCA [12] | 26.5 | 36.9 | 57.5 | 28.6 | 19.4 | 31.8 |
| MbLS [28] | 25.4 | 36.3 | 57.0 | 26.5 | 20.1 | 32.9 |
| TCD [33] | 25.2 | 36.4 | 57.2 | 26.8 | 19.1 | 31.5 |
| Cal-DETR (Ours) | 23.9 | 37.3 | 57.8 | 25.5 | 20.0 | 32.6 |

Table 2: *UP-DETR Calibration*: MS-COCO & CorCOCO. Results are reported on Epoch$_{150}$ setting and Cal-DETR shows improvement in calibration as compared to other methods including baseline.

find empirically over the validation set and use the logit mixing parameters $\alpha_1 = 0.9$, $\alpha_2 = 0.1$ and $\lambda = 0.5$ (see Sec. 4.2). For the classification loss, the focal loss is incorporated, and for localization, losses are generalized IoU and L1 as described in [26, 2]. For more implementation details, we refer readers to D-DETR [53], UP-DETR [5] and DINO [51].

## 4.1 Main Results

**Real and Corrupted:** We evaluate the calibration performance on large-scale datasets (MS-COCO as in-domain and its image corrupted version as out-domain, CorCOCO). The CorCOCO evaluation set is constructed by incorporating random severity and random corruption levels from [13]. Our method not only provides lower calibration errors but also shows improved detection performance. For D-DETR, in comparison to the baseline uncalibrated model, our method improves calibration of 4.4%$\downarrow$ (in-domain) and 1.9%$\downarrow$ (out-domain). In this dataset, we observe that classification-based train time calibration losses (MbLS [28] & MDCA [12]) are sub-optimal for the detection problems and worse than the baseline in some scenarios. We show that our method outperforms train time losses and achieves better calibration, e.g. in comparison with detection-based calibration loss (TCD [33]) improvements are 3.4%$\downarrow$ (in-domain) and 1.5%$\downarrow$ (out-domain). For UP-DETR, our method improves calibration from baseline by 1.6%$\downarrow$ (in-domain) and 2.0%$\downarrow$ (out-domain). For more results, we refer to Fig. 2, Tab. 1 & Tab. 2. We also provide the results on COCO (in-domain) and CorCOCO (out-domain) with the DINO model, as shown in Tab. 3. Our Cal-DETR improves the calibration performance of this strong baseline for both in-domain and out-domain settings.

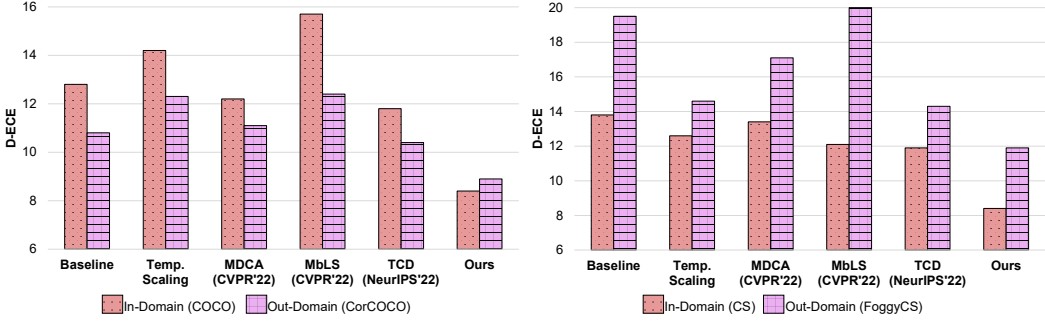

Figure 2: **D-DETR Calibration:** MS-COCO & Cityscapes as in-domains and respective Cor-COCO & Foggy Cityscapes as out-domains. Our proposed method (Cal-DETR) effectively decreases D-ECE compared to methods: baseline, post-hoc, and train-time losses.

| Methods | In-Domain (COCO) | | | Out-Domain (CorCOCO) | | |
|---|---|---|---|---|---|---|
| | D-ECE ↓ | AP ↑ | $AP_{0.5}$ ↑ | D-ECE ↓ | AP ↑ | $AP_{0.5}$ ↑ |
| Baseline$_{DINO}$ [51] | 15.5 | 49.0 | 66.6 | 13.2 | 27.3 | 39.2 |
| Temp. Scaling [10] | 15.1 | 49.0 | 66.6 | 14.3 | 27.3 | 39.2 |
| MDCA [12] | 16.3 | 48.6 | 66.1 | 14.0 | 26.7 | 38.4 |
| MbLS [28] | 19.1 | 48.6 | 66.0 | 15.9 | 26.9 | 38.4 |
| TCD [33] | 15.6 | 48.5 | 66.1 | 12.9 | 26.8 | 38.5 |
| Cal-DETR (Ours) | 11.7 | 49.0 | 66.5 | 10.6 | 27.5 | 39.3 |

Table 3: *DINO Calibration*: MS-COCO & CorCOCO. Results are reported on the Epoch12 (4-scale) setting and Cal-DETR shows improvement in calibration as compared to other methods including baseline.

Table 4: *D-DETR Calibration*: Results are shown in comparison with baseline, post-hoc, and train-time loss methods. Cal-DETR shows improvement in detection calibration on evaluation sets of the in-domain dataset (Cityscapes) and out-domain datasets (Foggy Cityscapes & BDD100k). AP and $AP_{0.5}$ are also reported.

| Methods | In-Domain (Cityscapes) | | | Out-Domain (Foggy Cityscapes) | | | Out-Domain (BDD100k) | | |
|---|---|---|---|---|---|---|---|---|---|
| | D-ECE ↓ | AP ↑ | $AP_{0.5}$ ↑ | D-ECE ↓ | AP ↑ | $AP_{0.5}$ ↑ | D-ECE ↓ | AP ↑ | $AP_{0.5}$ ↑ |
| Baseline$_{D-DETR}$ [53] | 13.8 | 26.8 | 49.5 | 19.5 | 17.3 | 29.3 | 11.7 | 10.2 | 21.9 |
| Temp. Scaling [10] | 12.6 | 26.8 | 49.5 | 14.6 | 17.3 | 29.3 | 24.5 | 10.2 | 21.9 |
| MDCA [12] | 13.4 | 27.5 | 49.5 | 17.1 | 17.7 | 30.3 | 14.2 | 10.7 | 22.7 |
| MbLS [28] | 12.1 | 27.3 | 49.7 | 20.0 | 17.1 | 29.1 | 11.6 | 10.5 | 22.7 |
| TCD [33] | 11.9 | 28.3 | 50.8 | 14.3 | 17.6 | 30.3 | 12.8 | 10.5 | 22.2 |
| Cal-DETR (Ours) | 8.4 | 28.4 | 51.4 | 11.9 | 17.6 | 29.8 | 11.4 | 11.1 | 23.9 |

**Weather:** Our method improves calibration over the baseline, post-hoc, and train-time losses with significant margins. Also, with our method, detection performance improves or is competitive with better calibration scores. Over baseline, we achieve an improvement of 5.4%↓ (in-domain) and 7.6%↓ (out-domain). Tab. 4 and Fig. 2 show further results.

**Urban Scene:** We show that our method improves calibration and/or competitive, especially in the out-domain. For example, with MDCA [12], we improve calibration by 5.0%↓ (in-domain) and 2.8%↓ (out-domain). Detection performance with our method is also improved. With comparison to [33], our method improves calibration by 1.4%↓. For more results, see Tab. 4.

**Synthetic and Real:** This scenario evaluates Sim10k as in-domain dataset and BDD100k subset reflecting same category with Sim10k as out-domain dataset. Over baseline, improvement of calibration for in-domain is 4.1%↓ and for out-domain 1.0%↓. Compared to MbLS [28], our method shows better calibration with the improvement of 16.3%↓ (in-domain) and 10.5%↓ (out-domain). Tab. 5 shows further results.

**Reliability diagrams:** Fig. 3 plots reliability diagrams on selected MS-COCO classes. It visualizes the D-ECE, and a perfect detection calibration means that precision and confidence are aligned.

## 4.2 Ablations & Analysis

We study ablation and sensitivity experiments with our proposed method. This section presents the impact of the components involved in our proposed method. For the choice of hyperparameters and sensitivity experiments, we split the training set of Cityscapes into train/validation sets (80%/20%).

| Methods | In-Domain (Sim10k) | | | Out-Domain (BDD100k) | | |
|---|---|---|---|---|---|---|
| | D-ECE ↓ | AP ↑ | $AP_{0.5}$ ↑ | D-ECE ↓ | AP ↑ | $AP_{0.5}$ ↑ |
| Baseline$_{D-DETR}$ [53] | 10.3 | 65.9 | 90.7 | 7.3 | 23.5 | 46.6 |
| Temp. Scaling [10] | 15.7 | 65.9 | 90.7 | 10.5 | 23.5 | 46.6 |
| MDCA [12] | 10.0 | 64.8 | 90.3 | 8.8 | 22.7 | 45.7 |
| MbLS [28] | 22.5 | 63.8 | 90.5 | 16.8 | 23.4 | 47.4 |
| TCD [33] | 8.6 | 66.4 | 90.4 | 6.6 | 23.1 | 46.0 |
| Cal-DETR (Ours) | 6.2 | 65.9 | 90.8 | 6.3 | 23.8 | 46.5 |

Table 5: *D-DETR Calibration*: Cal-DETR results in improved calibration performance over both in-domain and out-domain datasets, Sim10K and BDD100K respectively. The evaluation set contains car category. AP and $AP_{0.5}$ are also reported.

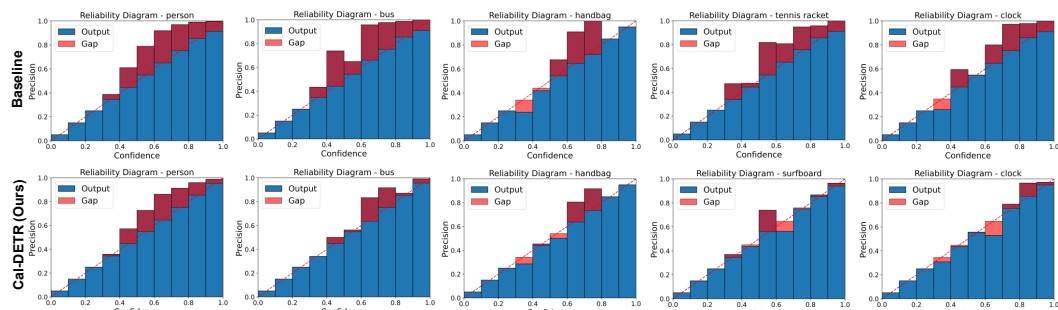

Figure 3: Reliability Diagrams: Baseline [53] vs. Cal-DETR (Ours): Reliability plots on MS-COCO dataset.

| Methods | | In-Domain (COCO) | | | Out-Domain (CorCOCO) | | |
|---|---|---|---|---|---|---|---|
| $L_{mod}$ | $L_{mix}$ | D-ECE $\downarrow$ | AP $\uparrow$ | $AP_{0.5} \uparrow$ | D-ECE $\downarrow$ | AP $\uparrow$ | $AP_{0.5} \uparrow$ |
| | | 12.8 | 44.0 | 62.9 | 10.8 | 23.9 | 35.8 |
| ✓ | | 11.0 | 44.1 | 62.8 | 9.9 | 23.6 | 35.5 |
| | ✓ | 9.5 | 43.5 | 63.3 | 9.2 | 23.3 | 35.3 |
| ✓ | ✓ | 8.4 | 44.4 | 64.2 | 8.9 | 24.0 | 36.3 |

Table 6: Impact of Components: Calibration results on MS-COCO & CorCOCO along with the detection performance are shown component-wise. Logit Modulation ($L_{mod}$) and Logit Mixing ($L_{mix}$).

Table 7: Impact of logit mixing and loss weightage with Cityscapes split (train/validation).

| Cal-DETR | In-Domain | | |
|---|---|---|---|
| | D-ECE $\downarrow$ | AP | $AP_{0.5}$ |
| $\alpha_1 = 0.9, \ \alpha_2 = 0.1$ | 8.1 | 25.0 | 46.8 |
| $\alpha_1 = 0.8, \ \alpha_2 = 0.2$ | 8.0 | 24.6 | 46.5 |
| $\alpha_1 = 0.7, \ \alpha_2 = 0.3$ | 10.1 | 25.0 | 46.7 |
| $\alpha_1 = 0.6, \ \alpha_2 = 0.4$ | 10.3 | 24.3 | 46.0 |

| Cal-DETR | In-Domain | | |
|---|---|---|---|
| | D-ECE $\downarrow$ | AP | $AP_{0.5}$ |
| $\lambda = 0.25$ | 10.4 | 24.6 | 46.1 |
| $\lambda = 0.325$ | 10.4 | 24.7 | 46.8 |
| $\lambda = 0.5$ | 8.1 | 25.0 | 46.8 |
| $\lambda = 0.75$ | 8.9 | 24.6 | 47.0 |
| $\lambda = 1.0$ | 8.7 | 24.4 | 46.4 |

(a) Impact of logit mixing $\alpha_1$ and $\alpha_2 = 1 - \alpha_1$ values in Cal-DETR. We observe improvement in detection performance by raising $\alpha$ values. We see that calibration performance is competitive with high $\alpha$ values.

(b) Loss weightage $\lambda$ values impact in Cal-DETR. We observe little improvement in detection performance (AP) on $\lambda = 0.5$ value. With the same value, we see that calibration performance is competitive.

**Impact of Components:** We separately investigate the function of logit modulation ($L_{mod}$) and logit mixing ($L_{mix}$) to see the impact on calibration performance. In Tab. 6, the calibration performance is improved over the baseline while incorporating individual components in our Cal-DETR. When we combine both, it further improves calibration performance along with the detection performance. We believe that this improvement could be due to greater diversity in logit space, since beyond mixed logits, we have access to non-mixed logits as well.

**Logit Mixing weightage:** We study the logit mixing weightage used in our proposed method on a separate validation set. Keeping the range of $\alpha_1 \in [0.6, 0.9]$, we observe in the experiments that the best trade-off of detection performance and model calibration is obtained at $\alpha_1 = 0.9$. We notice that calibration performance is competitive for higher $\alpha_1$ values but an overall improvement is observed for $\alpha_1 = 0.9$. Tab. 7a shows the behavior of $\alpha$ in Cal-DETR.

**Choice of single $\alpha$:** Several mixup strategies operate to mix an input image using another randomly selected image, so it usually involves two samples in the process. In our approach, we perform a query instance/object level mixup in the logit space. We first build a prototypical representation using all positive queries which is then used to achieve a mixup for a given positive query. Owing to this difference from conventional mixup strategies operating, our experiment with conventional mixup leads to suboptimal results. The subpar performance is potentially because the logit mixing suppresses the object representation with lower arbitrary $\alpha$ simultaneously having dominant prototypical representation from multiple positive queries. It is to be noted that the sampling strategy (Cal-DETR$_S$) still works better when compared to other calibration methods. We report the results in

| Methods | In-Domain (Cityscapes) | | | Out-Domain (Foggy Cityscapes) | | |
|---|---|---|---|---|---|---|
| | D-ECE ↓ | AP ↑ | $AP_{0.5}$ ↑ | D-ECE ↓ | AP ↑ | $AP_{0.5}$ ↑ |
| **Baseline $_{D\text{-}DETR}$ [53]** | 13.8 | 26.8 | 49.5 | 19.5 | 17.3 | 29.3 |
| **TCD [33]** | 11.9 | 28.3 | 50.8 | 14.3 | 17.6 | 30.3 |
| **Cal-DETR$_S$ (Ours)** | 11.7 | 28.2 | 52.3 | 13.2 | 17.3 | 29.6 |
| **Cal-DETR (Ours)** | 8.4 | 28.4 | 51.4 | 11.9 | 17.6 | 29.8 |

Table 8: Choice of singe $\alpha$: Cal-DETR outperforms its variant with sampling strategy (Cal-DETR$_S$) but it still works better when compared to other calibration methods.

the Tab. 8. We empirically find $\beta$ hyperparameter on beta distribution using the split of Cityscapes dataset. For our reported result, we choose $\beta$=0.3 using this procedure.

**Regularizer Loss weightage:** The impact of loss weightage is studied over the validation set, which shows that $\lambda = 0.5$ is a better choice to use with the regularizer function. Detection performance is increased with this setting along with the calibration performance. The trend shows that for a higher value of $\lambda$, calibration is more competitive (see Tab. 7b).

**D-UCE complements D-ECE:** To investigate the role of uncertainty calibration which complements confidence calibration, we analyze the detection expected uncertainty calibration error (D-UCE) [24] in Fig. 4. The formulation is described in the supplementary material.

**Limitation:** Our approach could struggle with the model calibration for long-tailed object detection and is a potential future direction.

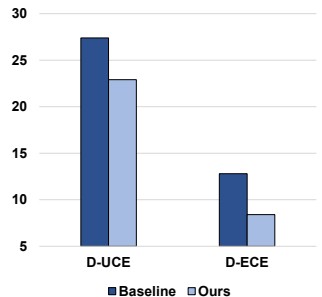

Figure 4: D-UCE for MS-COCO evaluation set. Uncertainty calibration error is reduced as compared to the baseline.

## 5   Conclusion

In this paper, we tackled the problem of calibrating recent transformer-based object detectors by introducing Cal-DETR. We proposed a new technique for estimating the uncertainty in transformer-based detector, which is then utilized to develop an uncertainty-guided logit modulation technique. To further enhance calibration performance, we developed a new logit mixing strategy which acts a regularizer on detections with task-specific loss. Comprehensive experiments on several in-domain and out-domain settings validate the effectiveness of our method against the existing train-time and post-hoc calibration methods. We hope that our work will encourage new research directions for calibrating object detectors.

## Acknowledgement

The computational resources were provided by the National Academic Infrastructure for Supercomputing in Sweden (NAISS), partially funded by the Swedish Research Council through grant agreement no. 2022-06725, and by the Berzelius resource, provided by the Knut and Alice Wallenberg Foundation at the National Supercomputer Center.

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
