*Supplementary Material*

# Cal-DETR: Calibrated Detection Transformer

**Muhammad Akhtar Munir**[1,2] , **Salman Khan**[1,3]**, Muhammad Haris Khan**[1],
**Mohsen Ali**[2]**, Fahad Shahbaz Khan**[1,4]
[1]Mohamed bin Zayed University of AI, [2]Information Technology University,
[3]Australian National University, [4]Linköping University
akhtar.munir@mbzuai.ac.ae

This supplementary material contains the formulation of the detection expected uncertainty calibration error (D-UCE) in Sec. 1 followed by quantitative results on the image corruption benchmark (Sec. 2). Then, we present the error bar plots with mean D-ECE and std deviation (Sec. 3). We also discuss some of the computing details (Sec. 4) followed by the qualitative results (Sec. 5).

## 1 D-UCE

Formulation of D-UCE is given by [3]:

$$
\text{D-UCE} = \sum_{b=1}^{B} \frac{|S(b)|}{|\mathcal{D}|} \left| \text{error}(b) - \text{uncertainty}(b) \right|,
\tag{1}
$$

where $\text{error}(b)$ is computed as an average error in a bin and $\text{uncertainty}(b)$ denotes the average uncertainty in a bin. $S(b)$ is the set of object instances in $b^{th}$ bin, and $|\mathcal{D}|$ contains the total number of samples. The error in particular detection is computed as it satisfies the false positive criteria.

## 2 Corruption Benchmark

Applying the image corruptions from [1] on MS-COCO with five different corruption types and a fixed severity level. We report D-ECE on these challenging out-domain scenarios. This further shows the effectiveness of our method to improve calibration in large-scale out-domain detection scenarios (Fig. 1).

## 3 Error Bars

We show the bar plots depicting mean D-ECE with respective standard deviations. Experiments have been executed using different seeds and in Cityscapes / Foggy Cityscapes scenarios. Fig. 2 shows the respective plots.

## 4 Compute Details

We provide computing details for our experiments. Our experiments have been performed under multi-GPU settings i.e. 4 GPUs (Tesla V100). Using this setting, we train the baselines (Deformable-DETR & UP-DETR) and the respective train-time calibration methods (MbLS, MDCA & TCD) on them, including our method (Cal-DETR). We evaluate D-ECE using the settings defined in [2].

37th Conference on Neural Information Processing Systems (NeurIPS 2023).

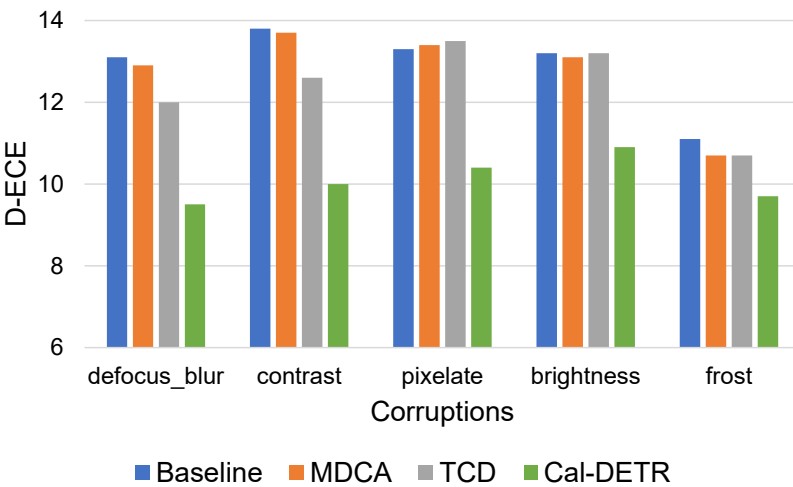

Figure 1: D-ECE↓ when Image Corruption are applied to MS-COCO dataset with five different corruption types under fixed severity level. Cal-DETR (Ours) shows improved calibration performance across different out-domain scenarios in comparison with baseline [4] and other train-time calibration loss methods.

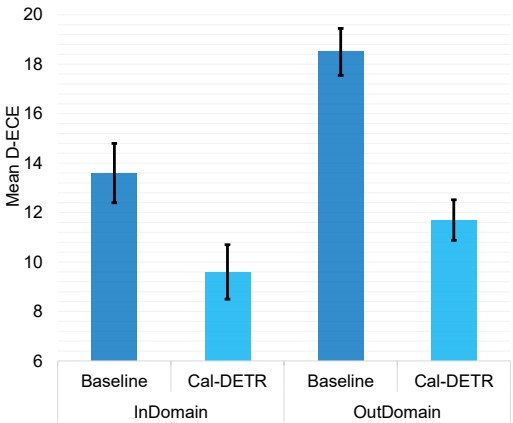

Figure 2: Mean D-ECE with std. deviation measures on Cityscapes (in-domain) and Foggy Cityscapes (out-domain). Experiments have been executed using different seeds. Cal-DETR (Ours) maintains the improvement in calibration while keeping less std deviation as compared to the baseline [4] for both in-domain and out-domain predictions.

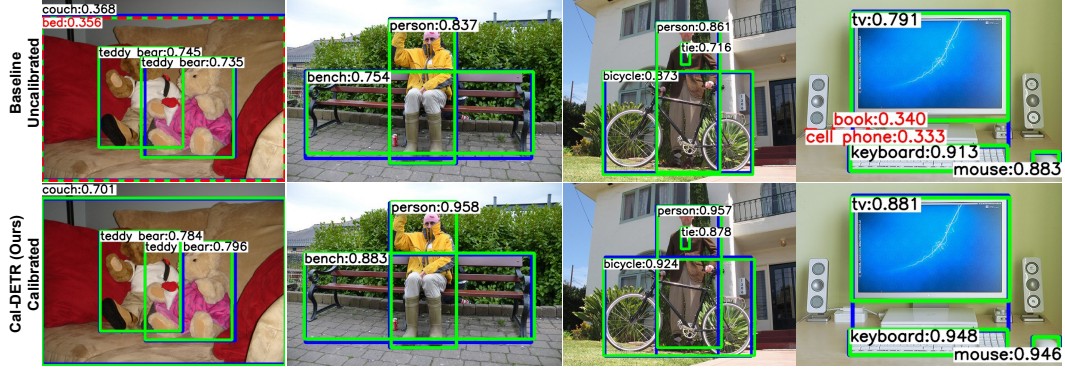

Figure 3: Qualitative: Baseline [4] vs. Cal-DETR (Ours): Results on MS-COCO dataset. The threshold set for results is 0.3. Green boxes are accurate predictions whereas Red (dashed) boxes are inaccurate predictions. Blue shows the ground truth boxes present for corresponding detections.

## 5 More Qualitative Results

In Fig. 3, we show additional qualitative results. It is observed that with the inclusion of our uncertainty-guided logit modulation and mixing modules, the model becomes more calibrated to predict accurate predictions with relatively good confidence scores.