# OpenReview forum: "Cal-DETR: Calibrated Detection Transformer"
_NeurIPS.cc/2023/Conference — NeurIPS 2023 poster_

### Official Review · Reviewer_Gc9J · 2023-06-29

**Soundness:** 3 good
**Presentation:** 3 good
**Contribution:** 3 good
**Rating:** 7
**Confidence:** 5

**Summary:**

The paper proposes a method to improve the calibration performance of transformer-based object detectors. In their approach, they first present a way to quantify the uncertainty of each logit using the variance of the outputs of different transformer decoder layers. Then, with the motivation that a higher uncertainty generally implies a poor calibration, they downscale the logits with high uncertainty. Second, they introduce a mixup based approach that is applied to the queries that match with the objects. Specifically, they create a prototype as the mean of the logits of the positive queries and mix it up with the logits of the foreground objects. They incorporate their method on Deformable DETR and UP-DETR, and observed notable improvements across different datasets.

**Strengths:**

- The proposed method is a training time approach, hence does not require an extra hold-out validation set.
- The improvement in the calibration performance is notable and outperforms existing approaches consistently in several datasets.
- While the reilabiility of the estimated uncertainties are not investigated, the proposed approach to do so is intuitive and does not introduce an additional burden on the detection architecture.
- The paper is written clearly and it is easy to follow.

**Weaknesses:**

- I think the logit mixup strategy that the authors introduce is quite related to a cited work [36]. The main differences are that the authors apply mixup in the logit space (which is again special case of manifold mixup), they use a fixed mixing coefficient instead of sampling it and the method is applied for object detection. I haven't seen in the paper (especially in related work or in Section 3.3.2) such a clarification. Therefore, I'd see the contribution of the authors in logit mixup as the extension of the regularized mixup [36] to object detection. I think this is still an important contribution but I'd like to see an explicit discussion with the prior related work.

- Recently [A] showed that the number of detections that are input to the calibration measure has a clear effect on the calibration performance. Specifically, such measures are easy to be misleading due to the maximum number of allowed detections, which is for example 100 for COCO dataset. For example, a detector that outputs 100 detection has more advantage to achieve lower calibration error compared to a detector that outputs less and does not fill the quota of 100. So, I wonder whether such a case exists here. Specifically for example for Table 1, how many detections per image on average do Baseline D-DETR, temp scaling, TCD (as the closest counterpart) and Cal-DETR output (which are then used as the inputs to estimate Detection ECE)? Or is there any thresholding of the top-100 detections while estimating the DECE? If so, how? Related to this, a minor suggestion (that I am not considering in my rating as a weakness as [A] came out in CVPR in June) can be to include a comparison in terms of Localisation-aware ECE in the way that [A] suggests to avoid such a doubt in the final version if the paper is accepted.

- The authors claim that they propose a method to quantify the uncertainty for each logit. However, further insight on these uncertainties are not provided. For example, there is no evidence that the estimated uncertainties are reliable and can be used for different purposes.

[A] Towards Building Self-Aware Object Detectors via Reliable Uncertainty Quantification and Calibration, CVPR 2023

**Questions:**

- Specifically for example for Table 1, how many detections per image on average do Baseline D-DETR, temp scaling, TCD (as the closest counterpart) and Cal-DETR output (which are then used as the inputs to estimate Detection ECE)? Or is there any thresholding of the top-100 detections while estimating the DECE?

- Can you please confirm that you obtain the prototypical representation query for each iteration during training? And I'd recommend making this explicit in the paper (L217-221). Also I'd recommend making it more explicit that you compute the mean of the logits across all positive queries (L217-218).

- In constrast to several mixup strategies that sample mixing coefficient (\alpha in Eq.(4)) from a distribution, why did you choose a single \alpha value?

- How do you obtain the labels after smoothing ( `c_i in L225). I think this should be explicitly defined.

**Limitations:**

Briefly mentioned

---

> ### Author Rebuttal · Authors · 2023-08-09
>
> **Q1: I think the logit mixup strategy that the authors introduce is quite related to a cited work [36]… I think this is still an important contribution but I'd like to see an explicit discussion...**
>
>  **A1:** Thanks for finding the logit mixup strategy an important contribution. Our approach can be seen as an extension of regularized mixup [36] from classification to object detection with important differences in the studied setting, the space in which mixup is applied, and the way mixup-based regularization is performed. We explain the differences and relationships between both approaches below:
>
> •	`High-level motivation:` In [36], the goal is to improve the uncertainty estimation of mixup without sacrificing accuracy, especially under severe distribution shifts for the task of image classification. The goal of our logit mixing is to further improve the calibration performance of DETR for the task of object detection (Cal-DETR).
>
> •	`Technical Details:` The mixup in [36] operates in input pixel space, studies the effect of hyperparameter beta of the sampling distribution for mixing coefficient and introduces a regularizer. Besides improving accuracy, it also helps in improving the quality of predictive uncertainty estimates. Ours operates in logit space, leverages positive queries to build a prototypical representation, and then uses this to achieve logit mixing for a given query. We also employ a regularizer loss, but it leverages the proposed logit mixing output. Our logit mixing complements the logit mixing and further improves the detection calibration performance of DETR-based pipelines.
>
> We will include the discussion with prior related work [36] in the final version.
>
> ---
>
> **Q2: In constrast to several mixup strategies that sample mixing coefficient (\alpha in Eq.(4)) from a distribution,...**
>
> **A2:** We empirically find $\beta$ hyperparameter on beta distribution using the split of Cityscapes dataset (L:298). For our reported result, we choose $\beta$=0.3 using this procedure.
>
> | $\beta$ (sampling) | D-ECE | AP   | mAP  |
> |-------------------|-------|------|------|
> | 0.3               | 11.4  | 24.5 | 46.4 |
> | 0.5               | 13.0  | 24.5 | 46.2 |
> | 1.0               | 13.6  | 24.7 | 45.7 |
>
> Several mixup strategies operate to mix an input image using another randomly selected image, so it involves two samples in the process. In our approach, we perform a query instance/object level mixup in the logit space. We first build a prototypical representation using all positive queries which is then used to achieve a mixup for a given positive query. Owing to this difference from conventional mixup strategies operating, our experiment with conventional mixup leads to suboptimal results.  The subpar performance is potentially because the logit mixing suppresses the object representation with lower arbitrary $\alpha$ simultaneously having dominant prototypical representation from multiple positive queries. It is to note that sampling strategy (Cal-DETR$_{sample}$) still works better when compared to other calibration methods. We report the results on Cityscapes and Foggy Cityscapes in the table below:
>
> |          		| In Domain |      |      	| Out Domain |      |      |
> |---------------------|-------|------|------|-------|------|------|
> | Methods             | D-ECE | AP   | mAP  | D-ECE | AP   | mAP  |
> | Baseline$_{D-DETR}$              | 13.8  | 26.8 | 49.5 	| 19.5  | 17.3 | 29.3 |
> | TS                  	    | 12.6  | 26.8 | 49.5 	| 14.6  | 17.3 | 29.3 |
> | MDCA                | 13.4  | 27.5 | 49.5 	| 17.1  | 17.7 | 30.3 |
> | MbLS                | 12.1  | 27.3 | 49.7 	| 20.0  | 17.1 | 29.1 |
> | TCD                  | 11.9  | 28.3 | 50.8 	| 14.3  | 17.6 | 30.3 |
> | **Cal-DETR$_{sample}$** | 11.7  | 28.2 | 52.3 	| 13.2  | 17.3 | 29.6 |
> | **Cal-DETR**         | 8.4   | 28.4 | 51.4 	| 11.9  | 17.6 | 29.8 |
>
> ---
>
> **Q3. Recently [A] showed that the number of detections that are…? Or is there any thresholding...?**
>
> **A3:** We thank the reviewer for mentioning useful insights about how the number of detections affects the calibration error. Note that in our implementation, we calibrate the detection model having prediction scores >=0.3. Specifically, we use a repository [22], where we set the threshold on scores prior to estimating the calibration which leads to a reliable and robust D-ECE measure in our case.
>
> ---
>
> **Q4. The authors claim that they propose a method to quantify the uncertainty for each logit. However,…**
>
> **A4:** [I] uses layer ensemble and some other works that use output space ensemble for the estimation of uncertainty (e.g., MC dropout/Deep Ensembles). In a similar spirit to these works, we exploit the logit space from different decoder layers (containing multiple dropout layers) for the estimation of uncertainty. Results show that our method is more effective in improving model calibration for both in-domain and out-domain scenarios.
>
> [I] Kushibar, Kaisar, et al. "Layer ensembles: A single-pass uncertainty estimation in deep learning for segmentation." International Conference on Medical Image Computing and Computer-Assisted Intervention, 2022.
>
> ---
>
> **Q5: Specifically for example for Table 1, how many detections… ? Or is there any thresholding...**
>
> **A5:** We explicitly use threshold on scores i.e. >=0.3 while estimating D-ECE
>
> ---
>
> **Q6: Can you please confirm that you obtain the prototypical representation query…**
>
> **A6:** Yes, we obtain the prototypical representation of queries for each iteration during training. We surely cater to the recommendations of the reviewer in the final version and add more details.
>
> ---
>
> **Q7: How do you obtain the labels after smoothing…**
>
> **A7:** $\alpha$ value for a given positive query contributes to smooth the label while remaining positive queries that contribute to forming a prototypical representation share 1-$\alpha$ value to smooth corresponding labels uniformly. We will add this detail in the final version.

---

> > ### Comment · Reviewer_Gc9J · 2023-08-20
> >
> > I thank the authors for their time and efforts to address my concerns. I am mostly satisfied with the rebuttal. My only point is: I do not agree that measuring the calibration error using 0.3 as a fixed confidence threshold provides a fair comparison. I can understand that this is the approach used in some repositories. However,
> >
> > - Reporting AP with top-100 detections (a potentially very low threshold) and DECE with 0.3 result in two configurations for object detector in a practical application. As a single setting can be used to get the detections in the application, I think this (using two settings, one for AP and one for DECE) is not a realistic way of comparing models.
> >
> > - A choice of 0.3 is also arbitrary. I am not sure if this threshold promotes or demotes some detectors.
> >
> > Just to make sure, if possible, could you please at least compute AP or LRP Errors of some models from 0.3 confidence score as well? This can make sure that the detector is not only calibrated but also it is accurate in the same setting compared to other methods.

---

> > > ### Author Response · Authors · 2023-08-20
> > >
> > > We thank you for reading our rebuttal and providing insightful comments. We report AP on top-100 detections, to avoid any confusion of performance drop when comparing with the standard settings. However, for the sake of completeness and as suggested, we report the AP/mAP on COCO (in-domain and out-domain) with same threshold (of 0.3) used for D-ECE. We observe that, even using the higher threshold, our Cal-DETR delivers the best detection performance with no drop across both in-domain and out-domain scenarios.
> > >
> > >
> > >
> > > |               | In Domain |      | Out Domain |      |
> > > |----------|-----------|------|------------|------|
> > > | Methods  | AP        | mAP  | AP         | mAP  |
> > > | Baseline$_{D-DETR}$   | 39.9      | 56.2 | 20.6       | 30.0 |
> > > | TCD          | 40.0      | 56.3 | 20.7       | 30.1 |
> > > | **Cal-DETR** | 40.6      | 57.9 | 21.1       | 31.3 |

---

> > > > ### Comment · Reviewer_Gc9J · 2023-08-20
> > > >
> > > > Thank you for your time. Given these results, I am happy to increase my score. I’d recommend the authors to consider additionally including in the camera ready version the following:
> > > >
> > > > - the methods evaluated as suggested in [A], i.e. using the best possible threshold for all models in terms of LRP Error and reporting LRP Error and DECE (and/or LaECE) with that confidence score threshold
> > > > - the AP and DECE from 0.30 threshold (as reported above)
> > > >
> > > > I believe this type of evaluation will also lead the following works on calibration properly and increase the value of this paper.

---

> > > > > ### Author Response · Authors · 2023-08-20
> > > > >
> > > > > We sincerely appreciate the constructive discussion. Thank you for the suggestions towards improving our submission.

---

### Official Review · Reviewer_co8a · 2023-07-05

**Soundness:** 3 good
**Presentation:** 3 good
**Contribution:** 3 good
**Rating:** 6
**Confidence:** 2

**Summary:**

This paper proposing a mechanism for calibrated detection transformers (Cal-DETR), particularly for Deformable-DETR and UP-DETR, which consists of quantifying uncertainty, an uncertainty-guided logit modulation and a logit mixing approach. Results show the method improves the baselines in calibrating both in-domain and out-domain detections while maintaining or even improving the detection performance.

**Strengths:**

I'm familiar with object detection and DETR, but not Model calibration.  It seems that the model calibration is a valuable problem, and the proposed method does improve it without performance drop.

**Weaknesses:**

It seems the performance of selected baselines is relatively low, e.g. the performance of Deformable-DETR is only 44.0 in COCO as mentioned in Table 1. Will the method still be effective when using a strong or even a sota model, like [DINO](https://github.com/IDEA-Research/DINO)?

**Questions:**

Will the method still be effective when using a strong or even a sota model?

**Limitations:**

The authors may try more baselines and report results.

---

> ### Author Rebuttal · Authors · 2023-08-09
>
> **Q1: Will the method still be effective when using a strong or even a sota model, like DINO?**
>
> **A1**: As suggested by the reviewer, we provide the results on COCO (in-domain) and CorCOCO (out-domain) with the DINO model, as shown below.  Our **Cal-DETR** improves the calibration performance of this strong baseline for both in-domain and out-domain settings.
>
> |          		| In Domain |      |      		| Out Domain |      |      |
> |----------|-----------|------|------|------------|------|------|
> | Methods  	| D-ECE     | AP   | mAP  	| D-ECE      | AP   | mAP  |
> | Baseline$_{DINO}$     	| 15.5      | 49.0 | 66.6 		| 13.2       | 27.3 | 39.2 |
> | TS       	| 15.1      | 49.0 | 66.6 		| 14.3       | 27.3 | 39.2 |
> | MbLS     	| 19.1      | 48.6 | 66.0 		| 15.9       | 26.9 | 38.4 |
> | MDCA     	| 16.3      | 48.6 | 66.1 		| 14.0       | 26.7 | 38.4 |
> | TCD      	| 15.6      | 48.5 | 66.1 		| 12.9       | 26.8 | 38.5 |
> | **Cal-DETR** 	| 11.7      | 49.0 | 66.5 		| 10.6       | 27.5 | 39.3 |
>
>
> DINO Calibration: MS-COCO & CorCOCO. Results are reported on the Epoch12 (4-scale) setting and Cal-DETR shows improvement in calibration as compared to other methods including baseline

---

> > ### Comment · Reviewer_co8a · 2023-08-18
> > **Seems good**
> >
> > I do not find any problem now and it seems a good work for me. Again, I must say that I'm not familiar with this field.

---

### Official Review · Reviewer_95V6 · 2023-07-07

**Soundness:** 2 fair
**Presentation:** 2 fair
**Contribution:** 2 fair
**Rating:** 5
**Confidence:** 3

**Summary:**

This paper focuses on performing calibration for DETR, particularly for Deformable-DETR and UP-DETR. The authors first propose an approach for quantifying uncertainty in DETRs, which is built from the variation in the output of decoder layers. Then they develop an uncertainty-guided logit modulation mechanism and a logit mixing approach as regularizers. Experiments on three in-domain and four out-domain scenarios show the effectiveness of the proposed method in calibrating.

**Strengths:**

The results of D-ECE are good in both in-domain and out-domain scenarios.
The experiments are extensive across various datasets and settings.

**Weaknesses:**

1. The proposed method has limited novelty. Although the authors claim that they do calibrate on object detection, the proposed method also performs the logits and shares large similarities with the methods in classification.
2. The way to quantify uncertainty in the DETR pipeline using the variation in the output of decoder layers is a straightforward idea, which may not be propriety to claim as a contribution. Moreover, the relationship between the proposed uncertainty-guided logit modulation mechanism and the logit mixing approach is not clear. It is more like a combination of two techniques, lacking deep insights.
3. When introducing D-ECE in Sec3.2, it would be better to cite [22] to indicate that the used measures follow previous works.

**Questions:**

1. I am confused that what benefits can we get when the D-ECE is lower. It seems that the box AP does not show clear differences between Cal-DETR and the baseline model in both in-domain and out-domain scenarios. If the calibration can not bring benefits in detecting objects, then what benefits can we get by performing calibration?
2. It seems that the proposed uncertainty-guided logit modulation and the logit mixing approach only perform on the output logits of the last decoder layer. Do the authors try to implement them into all decoder layers' output logits?
3. In Table 1 and Table 4, it seems that the D-ECE of the out-domain is lower than the D-ECE of the in-domain scenarios. Why? Does this phenomenon have a further explanation?

**Limitations:**

See the above sections.

---

> ### Author Rebuttal · Authors · 2023-08-09
>
> **Q1: The proposed method has limited novelty. Although the authors claim…**
>
> **A1:** To the best of our knowledge, this is the first work that strives to improve the calibration performance of recent SOTA ViT-based detectors by proposing an uncertainty-guided logit modulation and a logit mixing approach. We also refer to Reviewer Gc9J, who acknowledges the significance of our proposed approach, and Reviewer co8a appreciates no performance drop while improving model calibration. We would like to highlight that the relevant mixup strategies also target improved calibration, but majorly for the classification task [I, II], using image-level mixup in the input space. In our approach, we devise an uncertainty mechanism that does minimal changes to the model architecture and is also computationally efficient to get uncertainty-guided logits, which is followed by a logit mixing built with prototypical representations of positive queries. Both our methods are effective and complementary to each other in calibrating in-domain and out-domain predictions. Reviewer 95V6 comment does not provide details on what methods in classification lead to the concern about limited novelty. However, if specific methods are provided, we will be happy to explain the specific technical differences.
>
> [I] Zhang, Linjun, et al. "When and how mixup improves calibration." International Conference on Machine Learning. PMLR, 2022.
>
> [II] Thulasidasan, Sunil, et al. "On mixup training: Improved calibration and predictive uncertainty for deep neural networks." Advances in Neural Information Processing Systems 32 (2019).
>
> ------------------------------------------
>
> **Q2: The way to quantify uncertainty in the DETR pipeline using the variation in the output of decoder layers is a straightforward idea…**
>
> **A2:** Although quantifying uncertainty using variation in the output of decoder layers is a straightforward idea, and simple to implement, note that:
>
> `(a)`	It is effective for scaling the logits during train time to calibrate DETR-based object detectors both for in-domain and out-domain scenarios, as shown in our experiments.
>
> `(b)` Requires no modifications to the underlying architecture and is computationally efficient thereby revealing its plug-and-play nature (L:189-192).
>
> `(c)`	 Finally, to our knowledge, it has not been explored in quantifying uncertainty and calibrating modern DETR-based object detection pipelines.
>
> Akin to uncertainty-guided logit modulation, our logit mixing technique also operates in the logit space. It is designed to further enhance the calibration performance by introducing the logit mixing and a regularizer loss which allows model training using non-zero-entropy supervisory signal.
>
> ------------------------------------------
>
> **Q3: When introducing D-ECE in Sec3.2, it would be better to cite…**
>
> **A3:** We will add the reference [22] in Sec 3.2.
>
> ------------------------------------------
>
> **Q4: I am confused that what benefits can we get when the D-ECE is lower…**
>
> **A4:** The scope of model calibration is to reduce model miscalibration, which is quantified using the D-ECE metric while preserving the (object detection) performance. In other words, a calibrated model should be able to predict the class confidence that matches the actual likelihood of its correctness. It is of great value as this not only improve the overall trust in model predictions but could also increase the adoption of detector in several safety-critical applications (L:27-30).
>
> ------------------------------------------
>
> **Q5: It seems that the proposed uncertainty-guided logit modulation and the logit mixing approach only perform on the output logits of the last decoder layer…**
>
> **A5:** With minimal modification and keeping the DETR-based architecture intact, we implement the logit modulation and logit mixing to the last decoder layer only. DETR takes logits from the last layer only for probability output distribution. We provide results when uncertainty-guided logit modulation and logit mixing are applied to all decoder layers (**Cal-DETR$_{allDec}$**) on Cityscapes (in-domain) and Foggy Cityscapes (out-domain). The calibration performance of **Cal-DETR$_{allDec}$** is inferior to **Cal-DETR**, potentially due to less refined object queries (updates iteratively) as compared to the last layer.
>
>
> |          			| In Domain  |      |     	| Out Domain |      |      |
> |---------------------------|-------|------|------|-------|------|------|
> | Methods                   | D-ECE | AP   | mAP  | D-ECE | AP   | mAP  |
> | Baseline$_{D-DETR}$                    | 13.8  | 26.8 | 49.5 	| 19.5  | 17.3 | 29.3 |
> | TS                       	| 12.6  | 26.8 | 49.5 	| 14.6  | 17.3 | 29.3 |
> | MDCA                      | 13.4  | 27.5 | 49.5 	| 17.1  | 17.7 | 30.3 |
> | MbLS                      	| 12.1  | 27.3 | 49.7 	| 20.0  | 17.1 | 29.1 |
> | TCD                       	| 11.9  | 28.3 | 50.8 	| 14.3  | 17.6 | 30.3 |
> | **Cal-DETR$_{allDec}$** | 13.0  | 27.5 | 51.0 	| 12.8  | 17.3 | 29.7 |
> | **Cal-DETR**                 | 8.4   | 28.4 | 51.4 	| 11.9  | 17.6 | 29.8 |
>
> ------------------------------------------
>
> **Q6: In Table 1 and Table 4, it seems that the D-ECE of the out-domain…**
>
> **A6:** Modern deep neural networks are usually overconfident because they are trained with zero entropy signal with entropy minimization objective. We train the detection model with in-domain data only with modulation and logit mixing modules, that largely improves calibration for in-domain. Since our modules act as a regularizer for calibrating in-domain detections, it facilitates learning domain generalizable features [I], which also helps in calibrating out-domain detections and improving out-domain detection performance.
>
> [I] Wald, Yoav, et al. "On calibration and out-of-domain generalization." Advances in neural information processing systems 34 (2021): 2215-2227.

---

> > ### Author Response · Authors · 2023-08-18
> >
> > Dear Reviewer 95V6,
> >
> > Thanks again for your effort in reviewing our paper and giving us a helpful chance to improve the paper's quality. We hope that our response can address your concerns.
> >
> > Considering that the discussion period will end on August 21, we would like to know if you have any other questions about our paper, and we are glad to have a discussion with you in the following days. If our response has addressed your concerns, would you mind considering re-evaluating our work based on the updated information?
> >
> > Best regards,
> >
> > Authors

---

> > > ### Comment · Reviewer_95V6 · 2023-08-20
> > >
> > > Thanks for the authors' responses. I am not directly working on the model calibration, but part of my concerns are well addressed. I raised my score to borderline acceptance.

---

### Official Review · Reviewer_wX9Z · 2023-07-27

**Soundness:** 3 good
**Presentation:** 3 good
**Contribution:** 3 good
**Rating:** 5
**Confidence:** 4

**Summary:**

This paper proposes a calibrated detection transformer model, which equips the DETR variants with an uncertainty-guided logit modulator and a mixup augmentation for the classification branch of the detector. Specifically, the authors first quantify the uncertainty with the variance among the predicted logits from different decoder layers. Then logits with more uncertainty are suppressed. For mixup, it is similar to the common mixup used in detectors except the mix is between the average of the batch and each sample. Through experiments on various benchmarks, the authors demonstrate the effectiveness of the proposed method.

**Strengths:**

1. This paper studies an important but underexplored problem, calibration in object detectors.

2. The proposed method is lightweight and does not require any network modifications, which makes it more likely to be generalized to more architecture and application scenarios.

3. The authors perform very comprehensive experiments and the results show the effectiveness of the proposed method.

**Weaknesses:**

1. It is unclear why the variation in the output of decoder layers can measure the uncertainty of the class prediction. Is it based on intuition? And is there any experimental support?

2. As one of the two major contributions, logit mixing for confidence calibration is not technically novel enough. First, applying mixup for the classification branch of the object detector is considered a well-known technique [1][2]. Second, in [1], the mixup is considered a trick instead of a technical contribution. Finally, the difference in mixup between [1] and this paper is subtle.

[1] Zhang, Zhi, et al. "Bag of freebies for training object detection neural networks."
[2] Ge, Zheng, et al. "Yolox: Exceeding yolo series in 2021.

**Questions:**

Please refer to the weakness section.

---

> ### Author Rebuttal · Authors · 2023-08-09
>
> **Q1: It is unclear why the variation in the output of decoder layers can measure the uncertainty of the class prediction…**
>
> **A1:** Decoder layer contains multiple dropout layers that make it stochastic in nature and so allows capturing variation in logit space to estimate uncertainty. Logits are the raw output of the model, and with logits of all decoder layers, uncertainty is computed to modulate/scale logits of the final layer.
>
> ---
>
> **Q2: As one of the two major contributions, logit mixing for confidence calibration is not technically novel enough…**
>
> **A2:** `High-level intuition:` To our best of knowledge, our work is the first one that strives to improve the calibration performance of recent SOTA DETR-based pipelines by proposing an uncertainty-guided logit modulation and a logit mixing approach during training time without any hold out dataset (Reviewer Gc9J appreciates the significance of our approach). In [1], the aim is to apply complex spatial transforms to introduce occlusions and spatial signal perturbations in an effort to improve model generalization.  [2] applies the mixup from [1] and also leverages Mosaic as data augmentations for object detection to enhance detection performance. On the contrary, our main goal is to improve the calibration of  object detectors by developing a mixing approach in the logit space along with a regularizer loss that can be used with a detection-specific loss function.
>
> `Formulation and Technical Details:` [1] operates in input pixel space and performs a geometry preserved alignment of mixed images for object detection, where image pixels are mixed up and object labels are merged as a new array. Also, the rest of formulation details are same as the original mixup.  On the other hand, we operate in logit space, first, where we leverage positive queries to build a prototypical representation and then use this to achieve logit mixing for a given query. Furthermore, we employ a regularizer loss by leveraging the proposed logit mixing output. Overall, we fundamentally differ from [1] and [2] in terms of high-level intuition, formulation, and technical details.
>
> [1] Zhang, Zhi, et al. "Bag of freebies for training object detection neural networks."
>
> [2] Ge, Zheng, et al. "Yolox: Exceeding yolo series in 2021.

---

> > ### Comment · Reviewer_wX9Z · 2023-08-22
> >
> > Thanks for the detailed response. Most of my concerns are addressed and I decided to keep my initial rating (5 borderline accept).

---

### Author Rebuttal · Authors · 2023-08-10

We thank the reviewers (wX9Z, 95V6, co8a, Gc9J) for the positive and thoughtful feedback, and we appreciate the comments to improve our work.

**Reviewer Gc9J:** "The proposed method does not require an extra hold-out validation set, as it is a training time approach. Improvement in the calibration performance is notable. The proposed approach is intuitive and does not introduce an additional burden. Paper is easy to follow."

**Reviewer wX9Z:** "An important and underexplored problem, calibration in object detectors. The proposed method is lightweight and does not require any network modifications."

**Reviewer co8a:** "Model calibration is a valuable problem, and the proposed method does improve it without a performance drop."

**Reviewer 95V6:** "Calibration results are good in both in-domain and out-domain scenarios. Extensive experiments."

We summarize the main points presented in our response and we kindly hope that we have addressed all the questions.

+ We address the questions and provide clarifications and details that have improved our work.
+ We include comparisons with another DETR-based pipeline as well to show the effectiveness of our method.
+ We include more variants of our proposed approach that bring further insight into our choice.
+ We will update the final version based on the recommendations of the reviewers (i.e. additional results with a DETR-based DINO model to show scalability, provide insights and clarifications, and more discussions with close works in literature).

---

### Decision · Program_Chairs · 2023-09-21

**Decision:**

Accept (poster)

**Comment:**

The authors propose a method to calibrate DETR-based detection systems. The method applies to different variant (Deformable-DETR, DINO).
The authors addressed the concerns of all the reviewers during the rebuttal period, and no issues seem to remain as all reviewers are leaning towards acceptance.